# Transcriptome Analysis Reveals the Involvement of Alternative Splicing in the Nitrogen Starvation Response of *Chlamydomonas reinhardtii*

Xingcai Yang [1], Xiangyu Li [1,2], Jialin Zhao [1], Mingshi Xie [1], Xinyi Li [1], Bin Jia [1,*] and Ying Huang [1,*]

1 Guangdong Technology Research Center for Marine Algal Bioengineering, Guangdong Provincial Key Laboratory for Plant Epigenetics, Shenzhen Engineering Laboratory for Marine Algal Biotechnology, Longhua Innovation Institute for Biotechnology, College of Life Sciences and Oceanography, Shenzhen University, Shenzhen 518060, China
2 College of Physics and Optoelectronic Engineering, Shenzhen University, Shenzhen 518060, China
* Correspondence: jiabin@szu.edu.cn (B.J.); huangy@szu.edu.cn (Y.H.)

**Abstract:** Alternative splicing (AS) is a regulatory mechanism of post-transcriptional regulation that plays an important role in plant response to abiotic stresses. However, corresponding research involving the mechanism of AS in the nitrogen starvation response of *C. reinhardtii* is rare. This study performed a comprehensive and systematic analysis of AS events in *C. reinhardtii* at nine time points (0 h, 10 m, 30 m, 1 h, 6 h, 8 h, 24 h, and 48 h) under nitrogen starvation. It used STAR and rMATS tools to identify and quantify the probability of the AS event happening through the transcriptome high-throughput sequencing data. A total of 5806 AS events in 3500 genes were identified, and the retained intron and skipped exon were considered the main AS types. The genes related to the AS event in nitrogen starvation were mainly involved in spliceosome and transporter and enriched in the citrate cycle and fatty acid degradation pathways. These results suggested that AS may play an important role in the nitrogen starvation response in *C. reinhardtii*, and provided insights into post-transcriptional regulation under nitrogen starvation.

**Keywords:** alternative splice; abiotic stress; spliceosome; post-transcriptional regulation; *C. reinhardtii*





## 1. Introduction

Alternative splicing (AS) refers to the process of alternative splicing of a messenger ribonucleic acid (mRNA) precursor transcript into different mature mRNA transcripts, which is a necessary life process in eukaryotic gene post-transcriptional regulation [1]. Previous studies have classified AS events into the following five types based on the cross-linking of exons and introns on transcripts: skipped exon (SE), the alternative 3′ splice site (A3SS), the alternative 5′ splice site (A5SS), mutually exclusive exons (MXE), and retained intron (RI) [2–5]. AS occurs extensively in the transcription of eukaryotic genes; in *Homo sapiens*, *Arabidopsis thaliana*, and *Caenorhabditis elegans*, the proportion of genes with alternative splicing events was as high as 95%, 61%, and 71%, respectively [6–8].

The occurrence of AS can improve the diversity of proteins and the adaptability of organisms to the external environment, thus playing an essential role in plant growth and development, response to biotic and abiotic stresses, and evolution [9]. Plant genome studies have found that many mRNA transcripts have been shown to undergo AS in response to abiotic stress [10]. For example, the analysis of events in maize showed that drought stress induced tissue-dependent AS changes, which indicated that genes with AS changes were involved in the drought stress response of maize [11].

*C. reinhardtii* is a photosynthetic microalga with the advantages of a fast reproduction rate, high photosynthetic efficiency, and easy industrial cultivation, which has the potential to be used for the construction of algal strains for lipid production [12]. Previous studies

utilized genome-wide analysis of expressed sequence tag (EST) data and transcriptome data to identify the AS events in *C. reinhardtii*, and found that about 20% of genes (3342 out of 17,746 genes) undergo AS events in *C. reinhardtii* [13,14].

However, few studies have focused on the specific AS profile of *C. reinhardtii* under biotic or abiotic stresses. Nitrogen starvation, a common stress factor, can efficiently induce triacylglycerol (TAG) accumulation in *C. reinhardtii* [15]. During this process, complex pathways such as de novo fatty acid synthesis, membrane lipid remodeling, Kennedy Pathway, and β oxidation are reported to be involved [16–18]. There is extensive research on transcription regulation using transcriptome analysis in *C. reinhardtii* under nitrogen starvation, such as gene expression abundance [19], but few studies focus on post-transcriptional regulation. Genetic regulation mechanisms of microalgae are vital for screening potential strains with high levels of lipid production [20].

To identify the specific AS profile in the nitrogen starvation response of *C. reinhardtii*, this study performed a comprehensive and systematic analysis of AS events in *C. reinhardtii* at nine time points under nitrogen starvation. The genes related to AS events associated with nitrogen starvation were identified and subjected to Gene Ontology (GO) term and Kyoto Encyclopedia of Genes and Genomes (KEGG) pathway analyses. We found that RI was the main AS type and the genes of pathways related to nitrogen starvation undergo AS. These results suggested that AS may play an important role in the nitrogen starvation response in *C. reinhardtii*.

## 2. Materials and Methods

### 2.1. Transcriptome Data

The transcriptome data are from the SRA database of NCBI, and the accession number is PRJNA255778. The *C. reinhardtii* 4a+ wild-type algae strain was cultured at 25 °C under 100 με light irradiation in a liquid medium with 180 rpm to logarithmic phase, centrifuged and washed, and then transferred to nitrogen-free TAP. Sampling was performed at nine time points of 0 h, 10 m, 30 m, 1 h, 2 h, 6 h, 8 h, 24 h, and 48 h, and two duplicate samples were set at each time point, and then RNA extraction and library construction sequencing were performed [21]. This transcriptome experiment provides *C. reinhardtii* transcriptome samples at different time points after being transferred to nitrogen-free TAP, which can be used to identify AS events and analyze AS events changes in *C. reinhardtii* under nitrogen starvation.

### 2.2. Transcriptome Analysis

The SRAToolkit was used to download and decompress the transcriptome data. The study used fastp for quality control and data cleaning, including trimming adapter sequences and filtering low-quality reads [22]. Then, this study used the STAR for genome alignment with the two-pass alignment model, and the default parameters were used for other parameters [23]. The *C. reinhardtii* version 6.1 genome, which contributed to the sensitivity of AS event detection, was selected as the reference genome [24]. In this study, StringTie was used to quantify the gene expression abundance of transcriptome samples with TPM [25].

### 2.3. Gene Expression Correlation and Principal Component Analysis

Correlation analysis and principal component analysis of samples' gene expression were carried out using the R software, and the default parameters were used for function parameters.

### 2.4. Identification and Quantification of AS Events

The research used the software rMATs to identify AS events of different time points [3]. The junction counts of each identified AS event must be greater than 0 for each replicate sample. According to previous literature reports, strict testing threshold selection will improve the specificity of the detection of AS events [14].

In order to analyze the occurrence intensity of AS events, we designed ASLEVEL for quantification. The equation of ASLEVEL is shown in Formula (1).

$$\text{ASLEVEL} = \begin{cases} \text{mean(IncLevel)}, \text{AS} = \text{RI} \\ 1 - \text{mean(IncLevel)}, \text{AS} = \text{other} \end{cases} \tag{1}$$

IncLevel represents the ratio of the inclusion isoform to the total isoform in the repeated samples results of rMATS software, which refers to the level of inclusion isoform generated after the occurrence of an AS event in a transcript [3]. When the AS type is RI, the inclusion isoform represents the new isoform caused by the RI event. When the AS type is one of the other AS types, the inclusion isoform represents the original isoform produced when the AS event does not occur. ASLEVEL, through a simple adjustment of IncLevel, can be used to measure the proportion of new isoform produced after the occurrence of AS events, that is, the occurrence intensity of AS events or the probability of AS events happening.

### 2.5. Time-Series Cluster Analysis

The R package Mfuzz was used to perform the time-series clustering analysis on the ASLEVEL matrix of identified AS events [26]. Time series analysis is often used to analyze transcriptome gene expression trends to identify time-series-specific sets of expressed genes [27]. Similarly, time-series cluster analysis can determine the set of AS events that may be related to nitrogen starvation from all the identified AS events.

### 2.6. Gene Functional Annotation and Enrichment Analysis

To further determine the biological functions of AS events, biological function annotation and pathway enrichment analysis were performed on the genes related to AS events. Kyoto Encyclopedia of Genes and Genomes (KEGG) database [28] and Gene Ontology (GO) database [29,30] are used for gene function prediction and metabolic pathway prediction, respectively. The R package ClusterProfile was used to perform gene functional enrichment analysis [31]. In addition, Phytozome v13 was used for more detailed functional annotation of the AS-related genes [32].

### 2.7. Statistical Analysis

The enrichment significance was analyzed by Fisher's exact test. When $p$-value $\leq 0.05$, it was considered that there was significant enrichment of the GO term or KEGG pathway.

### 3. Results

### 3.1. Gene Expression Correlation and Principal Component Analysis

This study used TPM values to quantify the gene expression abundance of 18 transcriptome sequencing libraries under nitrogen starvation (Supplementary Tables S1 and S2). The gene expression correlation analysis showed that duplicate samples at different time points had a high correlation (Pearson correlation coefficient > 0.98). The correlation between samples gradually decreased with time points (Figure 1a), indicating that the samples of the dataset selected in the study had good repeatability. There were specific differences between different samples. Principal component analysis showed that repeated samples were centrally distributed by time point, and samples at different time points were sequentially distributed along the PC1 axis (Figure 1b), indicating that there were certain temporal differences in gene expression levels at different time points under nitrogen starvation.

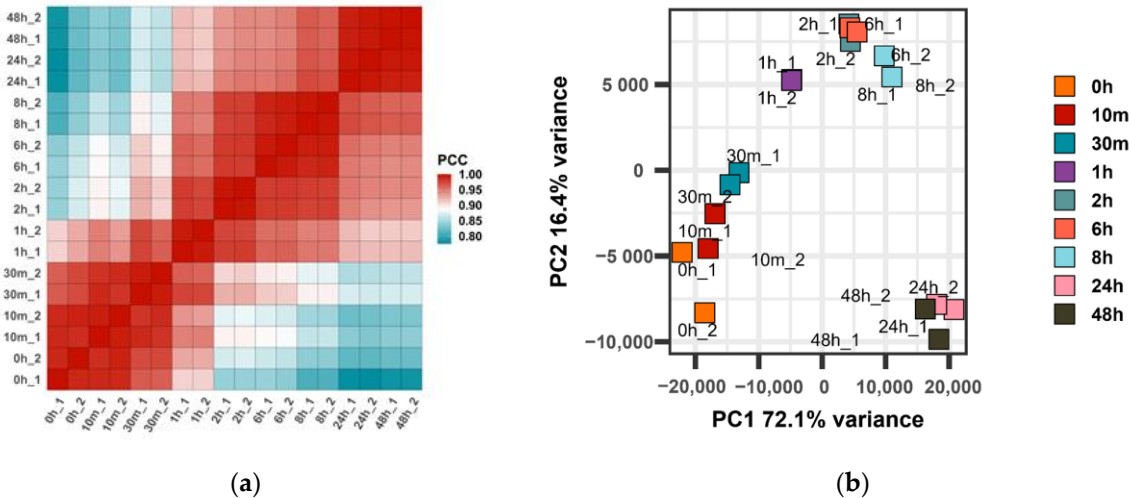

(**a**)　　　　　　　　　　　　　　　　(**b**)

**Figure 1.** Evaluation of transcriptome dataset used to identify alternative splicing (AS) events. (**a**) Gene Expression Correlation analysis of transcriptome samples. PCC: Pearson correlation coefficient. (**b**) Principal component analysis of transcriptome samples. PC1: Principal Component 1. PC2: Principal Component 2.

### 3.2. Identification of AS Events

This study used rMATS to identify the AS events of *C. reinhardtii*, which happened under nitrogen starvation. A total of 5806 AS events in 3500 genes were identified from the nine time points' samples. The count of AS events was calculated by event type, and the results showed that RI was the most frequent, followed by SE, A3SS, A5SS, and finally, MXE (Figure 2). Compared with other AS events, the number of detected SE events fluctuated wildly over time, and the difference between the maximum and the minimum number of SE events was 759. This may show a high sensitivity of SE events to nitrogen starvation. The highest total number of AS events were detected at one h under nitrogen starvation. The results showed that the same order of magnitude of alternative splicing events was detected in samples at different time points, which implies that AS events are prevalent at the transcriptome level. (Figure 2).

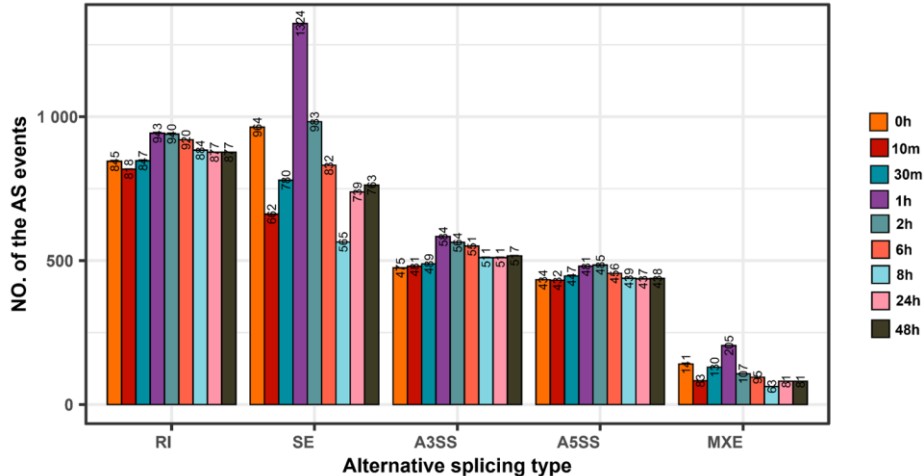

**Figure 2.** Statistics of the number of AS events in samples at different time points. RI was the most frequent, followed by SE, A3SS, A5SS, and MXE.

### 3.3. Quantification of AS Events and Time-Series Cluster Analysis

The study used ASLEVEL to quantify the probability of AS events happening, to find the AS events in response to nitrogen starvation (Supplementary Table S3). The ASLEVEL

can facilitate the trend analysis of the occurrence intensity of AS events. Time-Series Cluster Analysis was used to cluster all identified AS events by intensity trends. All AS events were clustered into six clusters. The time-series analysis showed that the ASLEVEL of Cluster 1 decreased significantly after 0 h, while the ASLEVEL of Cluster 2 increased with time points, inferring that Cluster 1 and 2 were related to nitrogen starvation. Thus, the AS events in Cluster 1 and Cluster 2 were classified as the nitrogen-starvation-responding AS events of *C. reinhardtii*, and the genes related to Cluster 1 and Cluster 2 were identified as AS-related genes (Figure 3).

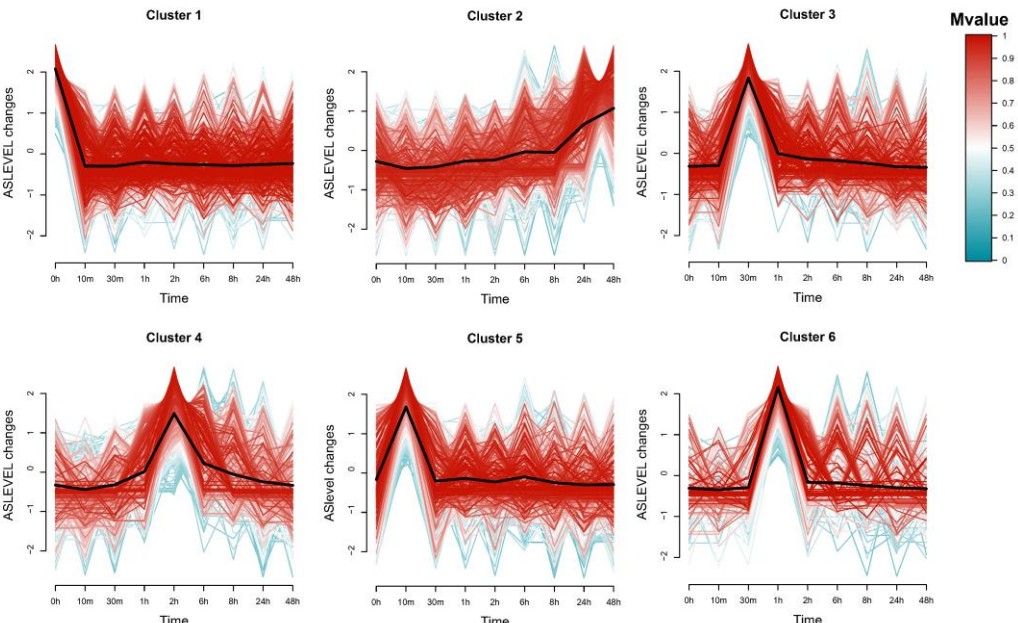

**Figure 3.** Results of Time-series Clustering. Different clusters represent AS events with different ASLEVEL change trends, respectively. Among them, Cluster 1 and Cluster 2 significantly decreased or increased ASLEVEL under nitrogen starvation. The black line of each cluster means the cluster centre for that cluster. Mvalue: Membership value.

*3.4. Functional Analysis of AS-Related Genes in Response to Nitrogen Starvation*

To determine how AS regulation affects the biological functions of *C. reinhardtii* in response to nitrogen starvation, we performed KEGG and GO enrichment analysis on the AS-related genes (Supplementary Tables S4 and S5). The KEGG functional enrichment analysis found that most genes were involved in transporters (ko02000), spliceosome (ko03041), DNA repair and recombination protein (ko03400), and protein kinases (ko01001). The result showed that the AS-related genes were enriched in the glyoxylate and dicarboxylate metabolism (ko00630, *p*-value = 0.00463), ABC transporters (ko02010, *p*-value = 0.00566), citrate cycle (TCA cycle, ko00020, *p*-value = 0.00726) and fatty acid degradation (ko00071, *p*-value = 0.01657). The genes also showed the enrichment effect in the pathways which involved protein and amino acid metabolisms, such as beta-Alanine metabolism (ko00410, *p*-value = 0.00842); valine, leucine, and isoleucine degradation (ko00280, *p*-value = 0.01374); and arginine biosynthesis (ko00220, *p*-value = 0.01374) (Figure 4a).

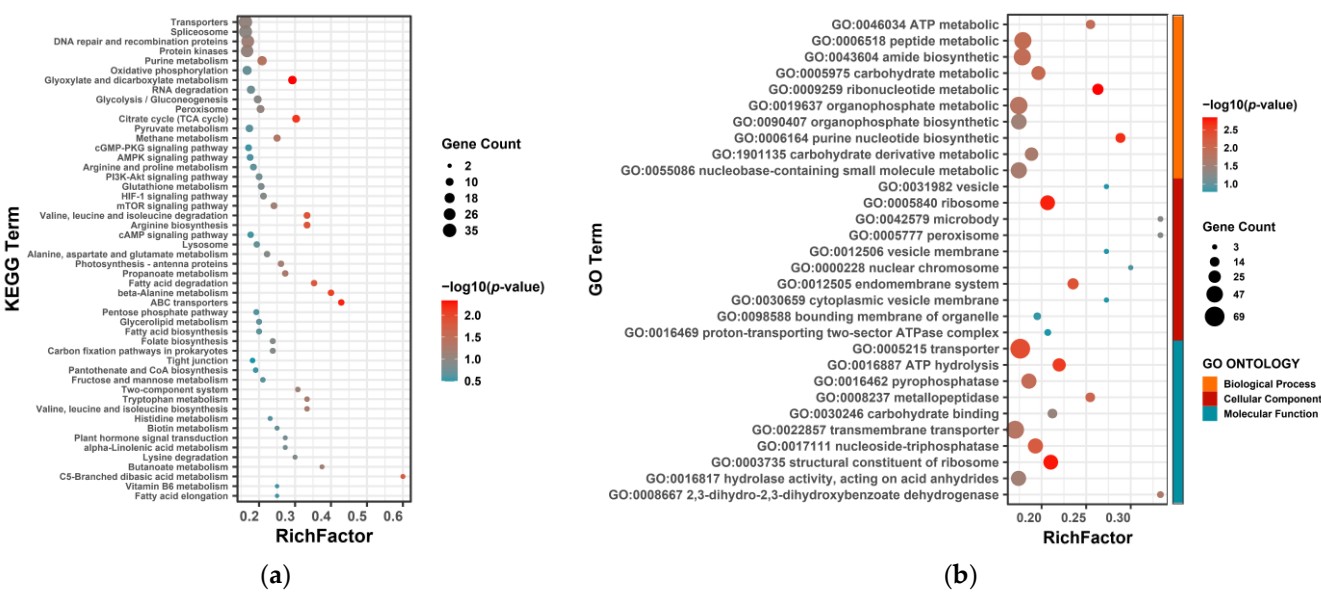

**Figure 4.** Functional analysis result. (**a**) KEGG enrichment result, which showed the top 50 terms of results; (**b**) GO enrichment, which shows the top 10 terms for each GO ontology.

GO ontology enrichment showed that the genes were involved in ribonucleotide-related metabolic processes (GO:0009259, GO:0019693, GO:0009260, GO:0046390), purine nucleotide-related metabolic processes (GO:0006164, GO:0009150, GO:0006163, GO:0009152), carbohydrate metabolic process (GO:0005975), and peptide related metabolic process (GO:0006518, GO:0043043). The genes were enriched in the ribosome (GO:0005840, *p*-value = 0.00175) and endomembrane system (GO:0012505, *p*-value = 0.00465) of cellular component ontology. Notably, AS-related genes showed higher enrichment factors on peroxisomes (GO:0005777). The genes were also enriched in structural constituent of ribosome (GO:0003735, *p*-value = 0.00163), ATP hydrolysis activity (GO:0016887, *p*-value = 0.00253), and transporter activity (GO:0005215, *p*-value = 0.00392) of molecular function ontology. The AS-related genes were enriched on carbohydrate binding (GO:0030246, *p*-value = 0.04000) and acyltransferase activity (GO:0016746, *p*-value = 0.04178) of molecular function ontology terms (Figure 4b).

### 3.5. Transcript Analysis of Genes Potentially Associated with Nitrogen Starvation

To further analyze the biological functions of AS-related genes, we annotated AS-related genes, identified AS types, and classified them according to metabolic pathway, signaling pathway, spliceosome, transporter, transcription factor, etc. (Supplementary Table S6). By analyzing the gene annotation results, we found that serine/arginine-rich pre-mRNA splicing factor (SRS1), SR-related pre-mRNA splicing factor (SRE1), phospho-glucose isomerase (PGI1), and type III polyketide synthase (PKS3) undergo AS events. The ASLEVEL of AS events of these genes was significantly increased under Nitrogen starvation, which may be caused by nitrogen stress (Figure 5a). Analysis of AS-related transcripts from SRS1, SRE1, PGI1, and PKS3 revealed that AS events in SRE1, PG1, and PKS3 caused changes in transcript CDS regions, thereby affecting the peptide chain of proteins. Among them, AS events occurring in PGI1 and PKS3 genes may cause premature terminator insertion (Figure 5b). The changes in transcripts of these genes suggested that these genes are undergoing post-transcription regulation. Previous transcriptome studies have found significant gene expression regulation in lipid-related metabolic pathways under Nitrogen starvation [33]. In addition, KEGG enrichment results showed that AS-related genes were significantly enriched in the citrate cycle (TCA cycle, ko00020, *p*-value = 0.00726) and fatty acid degradation (ko00071, *p*-value = 0.01657), which were related to lipid metabolic. Therefore, genes involved in lipid metabolic pathways were further analyzed. We found

that there are 36 AS-related genes that may be involved in the lipid-related metabolic pathway (Table 1).

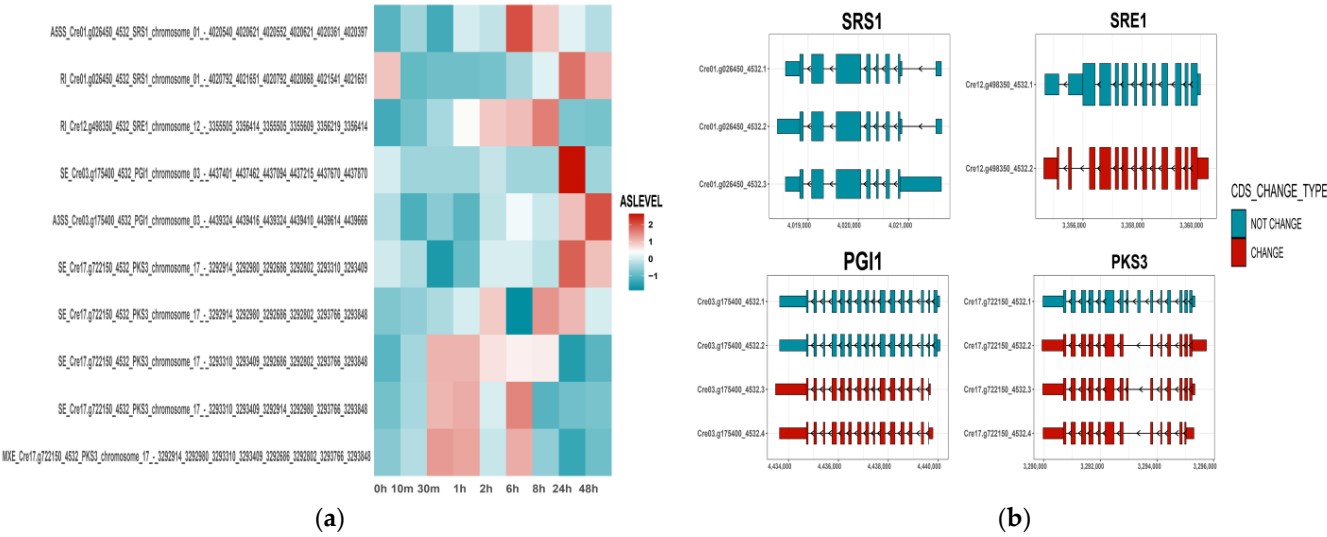

(**a**)                                           (**b**)

**Figure 5.** The change in ASLEVEL and CDS structure of Serine/arginine-rich pre-mRNA splicing factor (SRS1), SR-related Pre-mRNA splicing factor (SRE1), Phosphoglucose isomerase (PGI1), and Type III polyketide synthase (PKS3). (**a**) The ASLEVEL heatmap of AS event of SRS1, SRE1, PGI1, and PKS3. (**b**) The CDS structure change of SRS1, SRE1, PGI1, and PKS3. The narrow bars and wide bars represent the non-coding sequence and CDS structure of the transcript, respectively.

**Table 1.** Lipid-related metabolic pathway gene annotation and AS events.

| Gene_ID | Symbol | AS | Pfam_ID | Description | KEGG Pathway |
|---|---|---|---|---|---|
| Cre01.g005150_4532 | SGA1 | RI | PF00266 | Serine glyoxylate aminotransferase | Glyoxylate and dicarboxylate metabolism, Peroxisome |
| Cre01.g006950_4532 | FBA1 | A5SS | PF00274 | Fructose-1,6-bisphosphate aldolase | Glycolysis |
| Cre01.g042750_4532 | ACH1 | SE | PF00330, PF00694 | Aconitate hydratase | Glyoxylate and dicarboxylate metabolism, Citrate cycle |
| Cre02.g088250_4532 | MCT2 | A3SS | PF00698, PF12796, PF13857 | Malonyl-CoA:acyl-carrier-protein transacylase | Fatty acid biosynthesis |
| Cre02.g119000_4532 | PEX19 | RI | PF04614 | Putative peroxisome biogenesis protein | Peroxisome |
| Cre02.g141400_4532 | PCK1 | SE | PF01293 | Phosphoenolpyruvate carboxykinase | Citrate cycle, Glycolysis, Pyruvate metabolism |
| Cre02.g146050_4532 | ATO2 | SE | PF00108, PF00109, PF02803 | Acetyl-CoA acyltransferase | Glyoxylate and dicarboxylate metabolism, Pyruvate metabolism, Fatty acid degradation, Valine, leucine, and isoleucine degradation |
| Cre03.g144807_4532 | MAS1 | SE, MXE | PF01274 | Malate synthase | Glyoxylate and dicarboxylate metabolism, Pyruvate metabolism |
| Cre03.g149100_4532 | CIS2 | SE, MXE | PF00285 | Citrate synthase, glyoxysomal/microbody form | Glyoxylate and dicarboxylate metabolism, Citrate cycle |
| Cre03.g168700_4532 | NA | SE | PF13242 | Phosphoglycolate phosphatase/4-nitrophenylphosphatase | Glyoxylate and dicarboxylate metabolism |
| Cre03.g175400_4532 | PGI1 | A3SS,SE | PF00342 | Phosphoglucose isomerase 1 | Glycolysis |
| Cre03.g193850_4532 | SCLA1 | SE | PF00549, PF02629 | Succinyl-CoA ligase alpha chain | Citrate cycle |
| Cre03.g194850_4532 | MDH1 | A5SS | PF00056, PF02866 | NAD-dependent malate dehydrogenase 1, chloroplastic | Glyoxylate and dicarboxylate metabolism, Citrate cycle, Pyruvate metabolism |

**Table 1.** *Cont.*

| Gene_ID | Symbol | AS | Pfam_ID | Description | KEGG Pathway |
|---|---|---|---|---|---|
| Cre04.g214500_4532 | IDH3 | SE | PF00180 | Isocitrate dehydrogenase, NADP-dependent | Citrate cycle, Peroxisome |
| Cre06.g254400_4532 | FUM1 | SE | PF05681, PF05683 | Fumarate hydratase 1 | Citrate cycle, Pyruvate metabolism |
| Cre06.g262900_4532 | PFK1 | SE | PF00365 | Phosphofructokinase | Glycolysis |
| Cre06.g263602_4532 | PEX12 | MXE | PF04757 | Peroxin 12 | Peroxisome |
| Cre06.g272050_4532 | GPM1 | A3SS | PF01676, PF06415 | Phosphoglycerate mutase 1 | Glycolysis |
| Cre07.g343700_4532 | OGD2 | SE, MXE | PF00198, PF00364 | Dihydrolipoamide succinyltransferase, oxoglutarate dehydrogenase E2 component | Citrate cycle |
| Cre07.g347100_4532 | NA | SE | PF01263 | Putative sugar epimerase | Glycolysis |
| Cre07.g353450_4532 | ACS3 | A3SS, SE, MXE | PF00501, PF13193, PF16177 | Acetyl-CoA synthetase/ligase | Glyoxylate and dicarboxylate metabolism, Glycolysis, Pyruvate metabolism |
| Cre11.g467723_4532 | KAS1 | SE | PF00109, PF02801 | 3-ketoacyl-ACP-synthase | Fatty acid biosynthesis |
| Cre12.g483950_4532 | MDH4 | SE | PF00056, PF02866 | Malate dehydrogenase 4 | Glyoxylate and dicarboxylate metabolism, Citrate cycle, Pyruvate metabolism |
| Cre12.g500150_4532 | ALDH2 | SE | PF00171 | Aldehyde dehydrogenase | Glycolysis, Pyruvate metabolism, Fatty acid degradation, Valine, leucine, and isoleucine degradation |
| Cre12.g507400_4532 | LCS3 | SE | PF00501 | Long-chain acyl-CoA synthetase | Peroxisome, Fatty acid degradation, Fatty acid biosynthesis |
| Cre12.g510650_4532 | FBP2 | RI | PF00316 | Fructose-1,6-bisphosphatase, chloroplastic | Glycolysis |
| Cre12.g513200_4532 | ENO1 | A5SS, SE | PF00113, PF03952 | Enolase | Glycolysis |
| Cre12.g530600_4532 | GLN3 | A5SS | PF00120, PF03951 | Glutamine synthetase | Glyoxylate and dicarboxylate metabolism, Arginine biosynthesis |
| Cre12.g537200_4532 | OGD1 | SE | PF00676, PF02779, PF16078, PF16870 | 2-oxoglutarate dehydrogenase, E1 subunit | Citrate cycle |
| Cre15.g637761_4532 | NA | MXE | PF00005, PF06472 | Peroxisomal long-chain acyl-CoA transporter, ABC superfamily | Peroxisome |
| Cre16.g664550_4532 | SHMT1 | SE | PF00464 | Serine hydroxymethyltransferase | Glyoxylate and dicarboxylate metabolism |
| Cre16.g679200_4532 | PEX1 | SE | PF00004 | Peroxisome biogenesis protein | Peroxisome |
| Cre16.g689050_4532 | ACX1 | SE | PF01756, PF02770, PF14749 | Acyl-CoA oxidase/dehydrogenase | Peroxisome, Fatty acid degradation |
| Cre16.g695050_4532 | ECH3 | SE | PF00725, PF02737, PF16113, PF00378 | Enoyl-CoA hydratase 1 | Fatty acid degradation |
| Cre16.g695100_4532 | NA | SE | PF00441, PF02770, PF02771 | Putative Acyl-CoA oxidase | Peroxisome, Fatty acid degradation |
| Cre17.g722150_4532 | PKS3 | SE | PF02797, PF08392 | Type III polyketide synthase | Fatty acid elongation |

## 4. Discussion

The complete reference genome is conducive to comprehensively identifying AS events in the transcriptome data. Our study identified a total of 5806 AS events in 3500 genes from the transcriptome data of *C. reinhardtii* under nitrogen starvation. In addition, the previous study identified 3220 AS events in 2281 genes in the *C. reinhardtii* cell cycle transcriptome [14]. More AS events were identified in our study, which may be due to the more complete reference genome version v6.1, which has more transcript annotations than the previous version v5.6 (31862/19526) [24,34].

The RI and SE were found to be the main types of AS of *C. reinhardtii* under nitrogen starvation. These findings were consistent with the previous study, in which RI was the most common AS event (up to 40–50%) in *C. reinhardtii* [5,13]. Some studies also showed that RI was the most common AS type in the plant [35–37]. The number of SE events fluctuated with the change of time points, and recent research related to the analysis of AS in the drought response of *Glycyrrhiza uralensis* has shown that SE was the main type of AS [38], which indicates that SE events may be the most sensitive type of AS event in *C. reinhardtii* in response to nitrogen starvation.

The genes in the spliceosome and transporter pathways may be extensively regulated by AS events under nitrogen starvation. The KEGG result showed that AS-related genes associated with nitrogen starvation were mainly involved in the pathways of spliceosome and transporter. Spliceosomes are a dynamic family of particles that assemble on the mRNA precursor and remove noncoding introns [39]. The genes of the spliceosome undergo AS under nitrogen starvation, which may indicate that AS events were under the control of complex cross-regulation mechanisms that were largely consistent with the previous observation [40,41]. Some studies have shown that the transporter family has a tremendous structural diversity, and the AS event was considered to play an essential role in it [42,43].

The AS event may play an important role in the regulation of TAG accumulation in the nitrogen starvation response of *C. reinhardtii.* A previous study showed that nitrogen starvation enhances TAG accumulation in *C. reinhardtii* [44]. In addition, quantitative proteomic studies found that reducing citrate synthase in the citrate cycle would favor fatty acid biosynthesis, leading to an increase in TAG production [45]. The KEGG enrichment result showed that the AS-related gene was enriched significantly in the citrate cycle and fatty acid degradation, which suggests that AS may play a role in the down-regulation of the citrate cycle and fatty acid degradation pathway, and thus participate in TAG accumulation in the nitrogen starvation response of *C. reinhardtii.* It has been found that under nitrogen stress, the membrane lipids of *C. reinhardtii* is degraded to provide some fatty acids for the biosynthesis of TAG [12]. The AS-related genes were also enriched in peroxisome-related cellular components and the inner membrane system components of GO terms, which indicated that AS might relate to the remodeling of membrane lipids.

AS events probably take part in the regulation of protein degradation in the nitrogen starvation response of *C. reinhardtii*. In most microalgae with lipid accumulation, a decrease in protein content was observed under nitrogen stress, and one of the obvious reasons is that protein degradation can provide nitrogen elements required by microalgae to maintain cell life activities [46]. KEGG and GO analysis results showed that many AS-related genes were enriched in the metabolisms of amino acids and peptides. In addition, Studies have shown that *C. reinhardtii* will enter the gamete stage after nitrogen deprivation [47], and AS-related genes involved in nucleotide metabolism-related biological processes, DNA repair, and recombinant protein pathways which may indicate that the genes related cell cycle undergo AS [14].

This Study has found that AS also occurs in Serine/arginine-rich related genes, such as SRS1 and SRE1, in which RI events occur in SRE1, which may lead to changes in the CDS region of the SRE1 transcript and affect its protein structure. Previous studies have shown that the Serine/Arginine-rich protein family is related to the AS mechanism in post-transcriptional regulation in plants [48]. Further analysis of genes related to lipid metabolism pathways found that the occurrence of AS events could affect the CDS structure of PGI1 and PKS3, which may affect the stability of the transcript and may further affect the conformation of the functional domain of the protein [49]. PGI1 is involved in the glycolysis pathway and pyruvate synthesis [50], while PKS3 is involved in the fatty acid elongation pathway, which may be related to the change in fatty acid composition in the nitrogen starvation response of *C. reinhardtii* [17,18,51]. It was also found that glutamine synthetase (GLN3) undergoes a marked AS event under nitrogen starvation, which is accompanied by changes in gene expression. Glutamine synthetase is involved in nitrogen metabolic pathways, including amino acid synthesis and nitrogen uptake. The previous study shows that AS occurs on glutamine synthetase (OsGS1) of Oryza sativa, which generated two functional transcripts: OsGS1;1a and OsGS1;1b, which improved nitrogen use efficiency, affected grain development, and reduced amylose content [52].

## 5. Conclusions

This study has identified a set of AS events that were related to the nitrogen starvation response of *C. reinhardtii*. The results suggested that AS may play an important role in the regulation of genes related to TAG accumulation and protein degradation. This can be

used in future studies to explore the level of post-transcriptional regulation in *C. reinhardtii* under nitrogen starvation. In addition, AS events are widely involved in the regulation of spliceosome genes, implying that AS itself is regulated by complex cross-regulation mechanisms. In the future, it will be necessary to investigate the underlying mechanism in more detail.

**Supplementary Materials:** The following supporting information can be downloaded at: https://www.mdpi.com/article/10.3390/pr10122719/s1. Table S1: Gene Expression; Table S2: Transcript Expression; Table S3: ASLEVEL matrix of AS Events; Table S4: KEGG Enrichment Result; Table S5: GO Enrichment Result; Table S6: Annotation of the Genes.

**Author Contributions:** X.Y.: conceptualization, methodology, formal analysis, investigation, visualization, and writing original draft. X.L. (Xiangyu Li): resources, validation, and investigation. X.L. (Xinyi Li): formal analysis and investigation. J.Z.: resources and investigation. M.X.: resources and investigation. B.J. and Y.H.: writing—review and editing, supervision, and funding acquisition. All authors have read and agreed to the published version of the manuscript.

**Funding:** This research was funded by the China National Key Research and Development Project for Synthetic Biology (2018YFA0902500), the National Natural Science Foundation of China (31870343, 32273118), the Guangdong Basic and Applied Basic Research Foundation (2020A1515010352), Shenzhen Basic Research Projects (JCYJ20180507182405562), Shenzhen Special Fund for Sustainable Development (KCXFZ20211020164013021), the InnovationDriven Development Special Fund Project of Guangxi (Guike AA18242047), the Guangdong Natural Science Foundation (2020A1515010873), and R&D plan projects in key fields of Guangdong Province (2022B1111070005), and Shenzhen Stability support plan Projects(20220811124838001).

**Data Availability Statement:** All of the code files are available in GitHub repositories (https://github.com/156732531/AlternativeSplicingWorkflowForCrein, accessed on 14 November 2022).

**Conflicts of Interest:** The authors declare no conflict of interest.

## Abbreviations

The English abbreviations mentioned in the article and their meanings are listed below.

| | |
|---|---|
| AS | Alternative Splicing |
| TAP | Tris-Acetate-Phosphate |
| STAR | Spliced Transcripts Alignment to a Reference, the software used for genome alignment. (https://github.com/alexdobin/STAR) |
| StringTie | The software for Transcript assembly and quantification. |
| rMATS | Robust and Flexible Detection of Differential Alternative Splicing, the software used to identify alternative splicing events. (http://rnaseq-mats.sourceforge.net/, accessed on 14 November 2022) |
| mRNA | Messenger Ribonucleic Acid |
| SE | Skipped Exon |
| A3SS | the Alternative 3′ Splice Site |
| A5SS | the Alternative 5′ Splice Site |
| MXE | Mutually Exclusive Exons |
| RI | Retained Intron |
| EST | Expressed Sequence Tag |
| TAG | Triacylglycerol |
| SRA | Sequence Read Archive |
| NCBI | National Center for Biotechnology Information |
| SRAToolkit | The tools that download, manipulate and validate next-generation sequencing data stored in the NCBI SRA archive (https://github.com/ncbi/sra-tools, accessed on 14 November 2022) |
| Fastp | A tool designed to provide fast all-in-one preprocessing for FastQ files. (https://github.com/OpenGene/fastp, accessed on 14 November 2022) |
| TPM | Transcripts Per Kilobase per Million mapped reads |
| Mfuzz | the R package used to clustering |
| KEGG | Kyoto Encyclopedia of Genes and Genomes |

| | |
|---|---|
| GO | Gene Ontology |
| ClusterProfile | the R package used for gene functional enrichment analysis |
| Phytozome v13 | a comparative platform for green plant genomics (https://phytozome-next.jgi.doe.gov/, accessed on 14 November 2022) |
| PC1 | Principal Component 1 |
| PC2 | Principal Component 2 |
| PCC | Pearson Correlation Coefficient |
| Mvalue | Membership value |
| ABC | ATP binding cassette |
| TCA | Tricarboxylic Acid Cycle |
| SR | Serine/arginine-Rich |
| SRS1 | serine/arginine-rich pre-mRNA splicing factor |
| SRE1 | SR-related pre-mRNA splicing factor |
| PGI1 | phosphoglucose isomerase |
| PKS3 | type III polyketide synthase |
| CDS | Coding sequence |
| NAD | Nicotinamide Adenine Dinucleotide |
| OsGS | glutamine synthetase of Oryza sativa |
| GLN | glutamine synthetase |

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
