# Peer review of "Transcriptome Analysis Reveals the Involvement of Alternative Splicing in the Nitrogen Starvation Response of Chlamydomonas reinhardtii"

_processes, doi:10.3390/pr10122719_

Round 1

Reviewer 1 Report

It has been reported that lipid-producing algae Chlamydonas reinhardtii can accumulate a amount intracellular lipid-triacylglycerol under nitrogen starvation stress, which has an important application prospect. In this study, the authors studied the transcriptome regulation of Chlamydonas reinhardtii under nitrogen starvation stress, especially the regulation of Alternative splicing (AS) events in the transcriptome. The study is very meaningful and the authors have done a good job for the research. This study not only deepens the understanding of post-transcriptional regulation involving alternative splicing events in Chlamydonas reinhardtii, but also it can be very useful for a certain guidance in studies alternative splicing events in other eukaryotes.

Although the content and writing level of the manuscript are good, there are still some problems,which are listed below.

1. Most of the "nitrogen stress" mentioned in the manuscript uses N-condition except for the use of nitrogen stress or Nitrogen starvation in a few places. Is not appropriate.

2. In Section 3.2, the author divides alternative splicing events into five event types, namely RI, SE, A3SS, A5SS and MXE. What is the basis of this classification? All this should be explained in the manuscript.

3. It is mentioned in the Section "3.3 Quantification of AS Events and Time-Series Cluster Analysis" that there are six clusters. What do these clusters mean? Is it another category of alternative splicing events? So how is each Cluster divided? What genes does each cluster represent? Is it related to the "five alternative splicing event types" mentioned in Section 3.2?

4. The author did not give detailed interpretation to the results of some figures and table in the manuscript, such as Figure 3, Figure 5 and Table 1. The illustration content is too brief to be understood, especially what is the result of Figure 5? What does that mean? Also the Table 1 illustrates what the problem is.

5. Many English abbreviations appear in the manuscript, the full name should be indicate. But the author doesn't do that, which is bad for the reader's understanding of the paper.

6. Two specific incorrect words:

(1) Line 160, “for earch GO ontology” should be ”for each GO ontology”

(2) Line 170, “temrs” should be “terms”

Reviewer 2 Report

Thank you for your submission "Transcriptome analysis reveal the involvement of alternative splicing in Nitrogen starvation response of Chlamydonas reinhardtii". The manuscript is interesting but needs extensive editing before the consideration. Almost all the sections needs attention. Please find the attachment with comments.

Round 2

Reviewer 2 Report

Dear Authors

Thank you for submitting the revised version. The changes has been done extensively and manuscript is in better form now.